# GAUSSIAN CONDITIONAL RANDOM FIELDS FOR CLASSIFICATION

## ABSTRACT

In this paper, a Gaussian conditional random field model for structured binary classification (GCRFBC) is proposed. The model is applicable to classification problems with undirected graphs, intractable for standard classification CRFs. The model representation of GCRFBC is extended by latent variables which yield some appealing properties. Thanks to the GCRF latent structure, the model becomes tractable, efficient, and open to improvements previously applied to GCRF regression. Two different forms of the algorithm are presented: GCRF-BCb (GCRGBC - Bayesian) and GCRFBCnb (GCRFBC - non-Bayesian). The extended method of local variational approximation of sigmoid function is used for solving empirical Bayes in GCRFBCb variant, whereas MAP value of latent variables is the basis for learning and inference in the GCRFBCnb variant. The inference in GCRFBCb is solved by Newton-Cotes formulas for one-dimensional integration. Both models are evaluated on synthetic data and real-world data. It was shown that both models achieve better prediction performance than relevant baselines. Advantages and disadvantages of the proposed models are discussed.

## 1 INTRODUCTION

Increased quantity and variety of sources of data with correlated outputs, so called structured data, created an opportunity for exploiting additional information between dependent outputs to achieve better prediction performance. One of the most successful probabilistic models for structured output classification problems are conditional random fields (CRF) (Sutton & McCallum, 2006). The main advantages of CRFs lie in their discriminatory nature, resulting in the relaxation of independence assumptions and the label bias problem that are present in many graphical models. Aside of many advantages, CRFs also have many drawbacks mostly resulting in high computational cost or intractability of inference and learning. A wide range of different approaches of tackling these problems has been proposed, and they motivate our work, too.

One of the popular methods for structured regression based on CRFs – Gausian conditional random fields (GCRF) – has the form of multivariate Gaussian distribution (Radosavljevic et al., 2010). The main assumption of the model is that the relations between outputs are presented in quadratic form. It has convex loss function and, consequently, efficient inference and learning, and expensive sampling methods are not used.

In this paper, a new model of Gaussian conditional random fields for binary classification is proposed (GCRFBC). GCRFBC builds upon regression GCRF model which is used to define latent variables over which output dependencies are defined. The model assumes that discrete outputs $y_i$ are conditionally independent conditioned on continuous latent variables $z_i$ which follow a distribution modeled by a GCRF. That way, relations between discrete outputs are not expressed directly. Two different inference and learning approaches are proposed in this paper. The first one is based on evaluating empirical Bayes by marginalizing latent variables (GCRFBCb), whereas MAP value of latent variables is the basis for learning and inference in the second model (GCRFBCnb). In order to derive GCRFBCb model and its learning procedure the variational approximation of Bayesian logistic regression (Jaakkola & Jordan, 2000) is generalized.

Compared to CRFs and structured SVM classifiers, the GCRFBC models have some appealing properties:

- The model is applicable to classification problems with undirected graphs, intractable for standard classification CRFs. Thanks to the GCRF latent structure, the model becomes tractable, efficient and open to improvements previously applied to GCRF regression models.

- Defining correlations directly between discrete outputs may introduce unnecessary noise to the model (Tan et al., 2010). This problem can be solved by defining structured relations on a latent continuous variable space.

- In case that unstructured predictors are unreliable, which is signaled by their large variance (diagonal elements in the covariance matrix), it is simple to marginalize over latent variable space and obtain better results.

GCRFBC model is relying on the assumption that the underlying distribution of latent variables is multivariate normal distribution, due to that in the case when this distribution cannot be fitted well to the data (e.g. when the distribution of latent variables is multimodal) the model will not perform as well as it is expected. The proposed models are experimentally tested on both synthetic and real-world datasets in terms of predictive performance and computation time. In experiments with synthetic datasets, the results clearly indicate that the the empirical Bayes approach (GCRFBCb) better exploits output dependence structure, more so as the variance of the latent variables increases. We also tested both approaches on real-world datasets of predicting ski lift congestion, gene function classification, classification of music according to emotion and highway congestion. Both GCRFBC models outperformed ridge logistic regression, lasso logistic regression, neural network, random forest, and structured SVM classifiers, demonstrating that the proposed models can exploit output dependencies in a real-world setting.

## 2 RELATED WORK

An extensive review of binary and multi-label classification with structured output is provided in Su (2015). A number of different studies related to graph based methods for regression can be found in the literature (Fox, 2015). CRFs were successfully applied on a variety of different structured tasks (Cotterell & Duh, 2017; Zhang et al., 2015; Masada & Bunescu, 2017; Zia et al., 2018) and different model adaptations can be found in literature Kim (2017); Maaten et al. (2011). Recently, successful unifications of deep learning and CRFs have been proposed Chen et al. (2016); Kosov et al. (2018). Moreover, implementation of deep neural networks as potential functions is presented in form of structure prediction energy networks (SPEN) Belanger & McCallum (2016); Belanger et al. (2017). Adaptation of normalazing flows in SPEN structure is presented in Lu & Huang (2019).

An extensive review on topic of binary and multi-label classification with structured output is provided in Su (2015). Large number of different studies related to graph based methods for regression can be found in the literature (Fox, 2015). CRFs were successfully applied on a variety of different structured tasks, such as: low-resource named entity recognition (Cotterell & Duh, 2017), image segmentation (Zhang et al., 2015), chord recognition (Masada & Bunescu, 2017) and word segmentation (Zia et al., 2018). The mixture of CRFs capable to model data that come from multiple different sources or domains is presented in Kim (2017). The method is related to the well known hidden-unit CRF (HUCRF) (Maaten et al., 2011). The conditional likelihood and expectation minimization (EM) procedure for learning have been derived there. The mixtures of CRF models were implemented on several real-world applications resulting in prediction improvement. Recently, a model based on unification of deep learning and CRF was developed by Chen et al. (2016). The deep CRF model showed better performance compared to either shallow CRFs or deep learning methods on their own. Similarly, the combination of CRFs and deep convolutional neural networks was evaluated on an example of environmental microorganisms labeling (Kosov et al., 2018). The spatial relations among outputs were taken in consideration and experimental results have shown satisfactory results.

The GCRF model was first implemented for the task of low-level computer vision (Tappen et al., 2007). Since then, various different adaptations and approximations of GCRF were proposed (Radosavljevic et al., 2014). The parameter space for the GCRF model is extended to facilitate joint modelling of positive and negative influences (Glass et al., 2016). In addition, the model is extended by bias term into link weight and solved as a part of convex optimization. Semi-supervised

marginalized Gaussian conditional random fields (MGCRF) model for dealing with missing variables was proposed by Stojanovic et al. (2015). The benefits of the model were proved on partially observed data and showed better prediction performance than alternative semi-supervised structured models.A comprehensive review of continuous conditional random fields (CCRF) was provided in Radosavljevic et al. (2010). The sparse conditional random fields obtained by $l_1$ regularization are first proposed and evaluated by Wytock & Kolter (2013). Additionaly, Frot et al. (2018) presented GCRF with the latent variable decomposition and derived convergence bounds for the estimator that is well behaved in high dimensional regime. An adaptation of GCRF on discrete output was briefly discussed in Radosavljevic (2011), as a part of future work. This discussion motivates our work, but our approach is different in technical aspects.

## 3  METHODOLOGY

In this section we first present already known GCRF model for regression and then we propose GCRFBC model for binary classification and two approaches to inference and learning.

### 3.1  BACKGROUND MATERIAL

GCRF is a discriminative graph-based regression model (Radosavljevic et al., 2010). Nodes of the graph are variables $\boldsymbol{y} = (y_1, y_2, \ldots, y_N)$, which need to be predicted given a set of features $\boldsymbol{x}$. The attributes $\boldsymbol{x} = (\boldsymbol{x_1}, \boldsymbol{x_2}, \ldots, \boldsymbol{x_N})$ interact with each node $y_i$ independently of one another, while the relations between outputs are expressed by pairwise interaction function. In order to learn parameters of the model, a training set of vectors of attributes $x$ and real-valued response variables $y$ are provided. The generalized form of the conditional distribution $P\left(\boldsymbol{y}|\boldsymbol{x}, \boldsymbol{\alpha}, \boldsymbol{\beta}\right)$ is:

$$P\left(\boldsymbol{y}|\boldsymbol{x}, \boldsymbol{\alpha}, \boldsymbol{\beta}\right) = \frac{1}{Z\left(\boldsymbol{x}, \boldsymbol{\alpha}, \boldsymbol{\beta}\right)} \exp\left(-\sum_{i=1}^{N}\sum_{k=1}^{K}\alpha_k\left(y_i - R_k\left(\boldsymbol{x_i}\right)\right)^2 - \sum_{i \neq j}\sum_{l=1}^{L}\beta_l S_{ij}^l(y_i - y_j)^2\right) \tag{1}$$

First sum models relations between outputs $y_i$ and corresponding input vector $\boldsymbol{x_i}$ and the second one models pairwise relations between nodes. $R_k(\boldsymbol{x_i})$ represents an unstructured predictor of $y_i$ for each node in the graph and $S_{ij}^l$ is value that expresses similarity between nodes $i$ and $j$ in graph $l$. Unstructured predictor can be any regression model that gives prediction of output $y_i$ for given attributes $\boldsymbol{x_i}$. $K$ is the total number of unstructured predictors. $L$ is the total number of graphs (similarity functions). Graphs can express any kind of binary relations between nodes e.g., spatial and temporal correlations between outputs. $Z$ is a partition function and vectors $\boldsymbol{\alpha}$ and $\boldsymbol{\beta}$ are learnable parameters. One of the main advantages of GCRF is the ability to express different relations between outputs by variety of graphs and ability to learn which graphs are significant for prediction. The quadratic form of interaction and association potential enables conditional distribution $P(\boldsymbol{y}|\boldsymbol{x}, \boldsymbol{\alpha}, \boldsymbol{\beta})$ to be expressed as multivariate Gaussian distribution (Radosavljevic et al., 2010):

$$P(\boldsymbol{y}|\boldsymbol{x}, \boldsymbol{\alpha}, \boldsymbol{\beta}) = \frac{1}{(2\pi)^{\frac{N}{2}}|\Sigma|^{\frac{1}{2}}} \exp\left(-\frac{1}{2}(\boldsymbol{y} - \boldsymbol{\mu})^T \Sigma^{-1}(\boldsymbol{y} - \boldsymbol{\mu})\right) \tag{2}$$

Precision matrix $\Sigma^{-1} = 2Q$ and distribution mean $\boldsymbol{\mu} = \Sigma\boldsymbol{b}$ are defined as, respectively:

$$Q = \begin{cases} \sum_{k=1}^{K}\alpha_k + \sum_{h=1}^{N}\sum_{l=1}^{L}\beta_l S_{ih}^l, & \text{if } i = j \\ -\sum_{l=1}^{L}\beta_l S_{ij}^l, & \text{if } i \neq j \end{cases} \tag{3}$$

$$b_i = 2\left(\sum_{k=1}^{K}\alpha_k R_k(\boldsymbol{x_i})\right) \tag{4}$$

Due to concavity of multivariate Gaussian distribution, the inference task $\text{argmax}_{\boldsymbol{y}} P(\boldsymbol{y}|\boldsymbol{x}, \boldsymbol{\alpha}, \boldsymbol{\beta})$ is straightforward. The maximum posterior estimate of $\boldsymbol{y}$ is the distribution expectation $\boldsymbol{\mu}$.

The objective of the learning task is to optimize parameters $\boldsymbol{\alpha}$ and $\boldsymbol{\beta}$ by maximizing conditional log likelihood $\text{argmax}_{\boldsymbol{\alpha}, \boldsymbol{\beta}} \sum_{\boldsymbol{y}} \log P(\boldsymbol{y}|\boldsymbol{x}, \boldsymbol{\alpha}, \boldsymbol{\beta})$. One way to ensure positive definiteness of the covariance matrix of GCRF is to require diagonal dominance (Strang et al., 1993). This can be ensured by imposing constraints that all elements of $\boldsymbol{\alpha}$ and $\boldsymbol{\beta}$ be greater than 0 (Radosavljevic et al., 2010).

## 3.2 GCRFBC MODEL REPRESENTATION

One way of adapting GCRF to classification problem is by approximating discrete outputs by suitably defining continuous outputs. Namely, GCRF can provide dependence structure over continuous variables which can be passed through sigmoid function. That way the relationship between regression GCRF and classification GCRF is similar to the relationship between linear and logistic regression, but with dependent variables. Aside from allowing us to define a classification variant of GCRF, this may result in additional appealing properties: (i) The model is applicable to classification problems with undirected graphs, intractable for standard classification CRFs. Thanks to the GCRF latent structure, the model becomes tractable, efficient and open to improvements previously applied to GCRF regression models. (ii) Defining correlations directly between discrete outputs may introduce unnecessary noise to the model (Tan et al., 2010). We avoid this problem by defining structured relations on a latent continuous variable space. (iii) In case that unstructured predictors are unreliable, which is signaled by their large variance (diagonal elements in the covariance matrix), it is simple to marginalize over latent variable space and obtain better results.

It is assumed that $y_i$ are discrete binary outputs and $z_i$ are continuous latent variables assigned to each $y_i$. Each output $y_i$ is conditionally independent of the others, given $z_i$.

The conditional probability distribution $P(y_i|z_i)$ is defined as Bernoulli distribution:

$$P(y_i|z_i) = Ber(y_i|\sigma(z_i)) = \sigma(z_i)^{y_i}(1 - \sigma(z_i))^{1-y_i} \tag{5}$$

where $\sigma(\cdot)$ is sigmoid function. Due to conditional independence assumption, the joint distribution of outputs $y_i$ can be expressed as:

$$P(y_1, y_2, \ldots, y_N|\boldsymbol{z}) = \prod_{i=1}^{N} \sigma(z_i)^{y_i}(1 - \sigma(z_i))^{1-y_i} \tag{6}$$

Furthermore, the conditional distribution $P(\boldsymbol{z}|\boldsymbol{x})$ is the same as in the classical GCRF model and has canonical form defined by multivariate Gaussian distribution. Hence, joint distribution of continuous latent variables $\boldsymbol{z}$ and outputs $\boldsymbol{y}$ given $\boldsymbol{x}$ and $\boldsymbol{\theta} = (\alpha_1, \ldots, \alpha_K, \beta_1, \ldots, \beta_L)$ is is the general form of the GCRFBC model defined as:

$$P(\boldsymbol{y}, \boldsymbol{z}|\boldsymbol{x}, \boldsymbol{\theta}) = \prod_{i=1}^{N} \sigma(z_i)^{y_i}(1 - \sigma(z_i))^{1-y_i} \cdot \frac{1}{(2\pi)^{N/2} \left|\Sigma(\boldsymbol{x}, \boldsymbol{\theta})\right|^{1/2}}$$
$$\cdot \exp\left(-\frac{1}{2}(\boldsymbol{z} - \boldsymbol{\mu}(\boldsymbol{x}, \boldsymbol{\theta}))^T \Sigma^{-1}(\boldsymbol{x}, \boldsymbol{\theta})(\boldsymbol{z} - \boldsymbol{\mu}(\boldsymbol{x}, \boldsymbol{\theta}))\right) \tag{7}$$

We consider two ways of inference and learning in GCRFBC model: (i) GCRFBCb - with conditional probability distribution $P(\boldsymbol{y}|\boldsymbol{x}, \boldsymbol{\theta})$, in which variables $\boldsymbol{z}$ are marginalized over, and (ii) GCRFBCnb - with conditional probability distribution $P(\boldsymbol{y}|\boldsymbol{x}, \boldsymbol{\theta}, \mu_{\boldsymbol{z}})$, in which variables $\boldsymbol{z}$ are substituted by their expectations.

## 3.3 INFERENCE IN GCRFBCB MODEL

Prediction of discrete outputs $\boldsymbol{y}$ for given features $\boldsymbol{x}$ and parameters $\boldsymbol{\theta}$ is analytically intractable due to integration of the joint distribution $P(\boldsymbol{y}, \boldsymbol{z}|\boldsymbol{x}, \boldsymbol{\theta})$ with respect to latent variables. However, due to conditional independence between nodes, it is possible to obtain $P(y_i = 1|\boldsymbol{x}, \boldsymbol{\theta})$.

$$P(y_i = 1|\boldsymbol{x}, \boldsymbol{\theta}) = \int_{\boldsymbol{z}} \sigma(z_i) P(\boldsymbol{z}|\boldsymbol{x}, \boldsymbol{\theta}) d\boldsymbol{z} \tag{8}$$

where $\sigma(z_i)$ models $P(y_i|\boldsymbol{z})$. As a result of independence properties of the distribution, it holds $P(y_i = 1|\boldsymbol{z}) = P(y_i = 1|z_i)$, and it is possible to marginalize $P(\boldsymbol{z}|\boldsymbol{x}, \boldsymbol{\theta})$ with respect to latent variables $\boldsymbol{z}' = (z_1, \ldots, z_{i-1}, z_{i+1}, \ldots, z_N)$:

$$P(y_i = 1|\boldsymbol{x}, \boldsymbol{\theta}) = \int_{z_i} \sigma(z_i) \left(\int_{\boldsymbol{z}'} P(\boldsymbol{z}', z_i|\boldsymbol{x}, \boldsymbol{\theta}) d\boldsymbol{z}'\right) dz_i \tag{9}$$

where $\int_{z'} P(z', z_i | x, \theta) dz'$ is normal distribution with mean $\mu = \mu_i$ and variance $\sigma_i^2 = \Sigma_{ii}$. Therefore, it holds:

$$P(y_i = 1 | x, \theta) = \int_{-\infty}^{+\infty} \sigma(z_i) \mathcal{N}(z_i | \mu_i, \sigma_i^2) dz_i \tag{10}$$

The evaluation of $P(y_i = 0 | x, \theta)$ is straightforward: $P(y_i = 0 | x, \theta) = 1 - P(y_i = 1 | x, \theta)$.

The one-dimensional integral is still analytically intractable, but can be effectively evaluated by one-dimensional numerical integration. The proposed inference approach can be effectively used in case of huge number of nodes, due to low computational cost of one-dimensional numerical integration.

### 3.4 INFERENCE IN GCRFBCnb MODEL

The inference procedure in GCRFBCnb is much simpler, because marginalization with respect to latent variables is not performed. To predict $y$, it is necessary to evaluate posterior maximum of latent variable $z_{\max} = \underset{z}{\operatorname{argmax}} P(z | x, \theta)$, which is straightforward due to normal form of GCRF. Therefore, it holds $z_{\max} = \mu_{z,i}$. The conditional distribution $P(y_i = 1 | x, \mu_{z,i}, \theta)$, where $\mu_{z,i}$ is expectation of latent variable $z_i$, can be expressed as:

$$P(y_i = 1 | x, \mu_z, \theta) = \sigma(\mu_{z,i}) = \frac{1}{1 + \exp(-\mu_{z,i})} \tag{11}$$

### 3.5 LEARNING IN GCRFBCb MODEL

In comparison with inference, learning procedure is more complicated. Evaluation of the conditional log likelihood is intractable, since latent variables cannot be analytically marginalized. The conditional log likelihood is expressed as:

$$\mathcal{L}(Y | X, \theta) = \log \int_Z P(Y, Z | X, \theta) dZ = \sum_{j=1}^M \log \int_{z_j} P(y_j, z_j | x_j, \theta) dz_j = \sum_{j=1}^M \mathcal{L}_j(y_j | x_j, \theta) \tag{12}$$

$$\mathcal{L}_j(y_j | x_j, \theta) = \log \int_{z_j} \prod_{i=1}^N \sigma(z_{ji})^{y_{ji}} (1 - \sigma(z_{ji}))^{1-y_{ji}} \frac{\exp(-\frac{1}{2}(z_j - \mu_j)^T \Sigma_j^{-1}(z_j - \mu_j))}{(2\pi)^{N/2} |\Sigma_j|^{1/2}} dz_j \tag{13}$$

where $Y \in \mathbb{R}^{M \times N}$ is complete dataset of outputs, $X \in \mathbb{R}^{M \times N \times A}$ is complete dataset of features, $M$ is the total number of instances and $A$ is the total number of features. Please note that each instance is structured, so while different instances are independent of each other, variables within one instance are dependent.

One way to approximate integral in conditional log likelihood is by local variational approximation. Jaakkola & Jordan (2000) derived lower bound for sigmoid function, which can be expressed as:

$$\sigma(x) \geqslant \sigma(\xi) \exp\{(x - \xi)/2 - \lambda(\xi)(x^2 - \xi^2)\} \tag{14}$$

where $\lambda(\xi) = -\frac{1}{2\xi} \cdot \left[\sigma(\xi) - \frac{1}{2}\right]$ and $\xi$ is a variational parameter. The Eq. 14 is called $\xi$ *transformation* of sigmoid function and it yields maximum value when $\xi = x$. This approximation can be applied to the model defined by Eq. 13, but the variational approximation has to be further extended because of the product of sigmoid functions, such that:

$$P(y_j, z_j | x_j, \theta) = P(y_j | z_j) P(z_j | x_j, \theta) \geq \underline{P}(y_j, z_j | x_j, \theta, \xi_j) \tag{15}$$

$$\underline{P}(y_j, z_j | x_j, \theta, \xi_j) = \prod_{i=1}^N \sigma(\xi_{ji}) \exp\left(z_{ji} y_{ji} - \frac{z_{ji} + \xi_{ji}}{2} - \lambda(\xi_{ji})(z_{ji}^2 - \xi_{ji}^2)\right) \cdot$$
$$\frac{1}{(2\pi)^{N/2} |\Sigma_j|^{1/2}} \exp\left(-\frac{1}{2}(z_j - \mu_j)^T \Sigma_j^{-1}(z_j - \mu_j)\right) \tag{16}$$

The Eq. 16 can be arranged in the form suitable for integration. Detailed derivation of lower bound of conditional log likelihood is presented in Appendix A. The lower bound of conditional log likelihood

$\underline{\mathcal{L}}(\boldsymbol{y_j}|\boldsymbol{x_j}, \boldsymbol{\theta}, \boldsymbol{\xi_j})$ is defined as:

$$\underline{\mathcal{L}}_j(\boldsymbol{y_j}|\boldsymbol{x_j}, \boldsymbol{\theta}, \boldsymbol{\xi_j}) = \log \underline{P}(\boldsymbol{y_j}|\boldsymbol{x_j}, \boldsymbol{\theta}, \boldsymbol{\xi_j}) = \sum_{i=1}^{N} \left( \log \sigma(\xi_{ji}) - \frac{\xi_{ji}}{2} + \lambda(\xi_{ji}) \xi_{ji}^2 \right) -$$
$$\frac{1}{2} \boldsymbol{\mu_j^T} \Sigma_j^{-1} \boldsymbol{\mu_j} + \frac{1}{2} \boldsymbol{m_j^T} S_j^{-1} \boldsymbol{m_j} + \frac{1}{2} \log |S_j| \quad (17)$$

where:

$$S_j^{-1} = \Sigma_j^{-1} + 2\Lambda_j \qquad \boldsymbol{m_j} = \Sigma_j \left( \left( \boldsymbol{y_j} - \frac{1}{2} \boldsymbol{I} \right) + \Sigma_j^{-1} \boldsymbol{\mu_j} \right) \quad (18)$$

$$\Lambda_j = \begin{bmatrix} \lambda(\xi_{j1}) & 0 & 0 & \dots & 0 \\ 0 & \lambda(\xi_{j2}) & 0 & \dots & 0 \\ \vdots & \vdots & \vdots & \ddots & \vdots \\ 0 & 0 & 0 & \dots & \lambda(\xi_{jN}) \end{bmatrix} \quad (19)$$

GCRFBCb uses the derivative of conditional log likelihood in order to find the optimal values for parameters $\boldsymbol{\alpha}$, $\boldsymbol{\beta}$ and matrix of variational parameters $\boldsymbol{\xi} \in \mathbb{R}^{M \times N}$. In order to ensure positive definiteness of normal distribution involved, it is sufficient to constrain parameteres $\boldsymbol{\alpha} > 0$ and $\boldsymbol{\beta} > 0$. The partial derivatives of lower bound of conditional log likelihood are presented in Appendix B. For constrained optimization, the truncated Newton algorithm was used Nocedal & Wright (2006); Facchinei et al. (2002). The target function is not convex, so finding a global optimum cannot be guaranteed.

## 3.6 LEARNING IN GCRFBCNB MODEL

In GCRFBCnb the mode of posterior distribution of continuous latent variable $\boldsymbol{z}$ is evaluated directly, so there is no need for approximation. The conditional log likelihood can be expressed as:

$$\mathcal{L}\left(\boldsymbol{Y}|\boldsymbol{X}, \boldsymbol{\theta}, \boldsymbol{\mu}\right) = \log P(\boldsymbol{Y}|\boldsymbol{X}, \boldsymbol{\theta}, \boldsymbol{\mu}) = \sum_{j=1}^{M} \sum_{i=1}^{N} \log P(y_{ji}|\boldsymbol{x_j}, \boldsymbol{\theta}, \mu_{ji}) = \sum_{j=1}^{M} \sum_{i=1}^{N} \underline{\mathcal{L}}_{ji}(y_{ji}|\boldsymbol{x_j}, \boldsymbol{\theta}, \mu_{ji})$$
$$(20)$$
$$\mathcal{L}_{ji}(y_{ji}|\boldsymbol{x_j}, \boldsymbol{\theta}, \mu_{ji}) = y_{ji} \log \sigma(\mu_{ji}) + (1 - y_{ji}) \log \left( 1 - \sigma(\mu_{ji}) \right) \quad (21)$$

The partial derivatives of conditional log likelihood are presented in Appendix C.

## 4 EXPERIMENTAL EVALUATION

Both proposed models were tested and compared on synthetic data and real-world tasks.[1] All compared classifiers were compared in terms of the area under ROC curve (AUC) and accuracy [2] (ACC). Moreover, the lower bound (in case of GCRFBCb) of conditional log likelihood $\underline{\mathcal{L}}\left(\boldsymbol{Y}|\boldsymbol{X}, \boldsymbol{\theta}, \boldsymbol{\mu}\right)$ and actual value (in case of GCRFBCnb) of conditional log likelihood $\mathcal{L}\left(\boldsymbol{Y}|\boldsymbol{X}, \boldsymbol{\theta}\right)$ of obtained values on synthetic test dataset were also reported.

## 4.1 SYNTHETIC DATASET

The main goal of experiments on synthetic datasets was to examine models under various controlled conditions, and show advantages and disadvantages of each. In all experiments on synthetic datasets two different graphs were used (hence $\boldsymbol{\beta} \in \mathbb{R}^2$) and two unstructured predictors (hence $\boldsymbol{\alpha} \in \mathbb{R}^2$). The results of experiments on synthetic datasets are presented in Appendix D.

It can be noticed, that in cases where norm of the variances of latent variables is small, both models have equal performance considering AUC and conditional log likelihood $\underline{\mathcal{L}}\left(\boldsymbol{Y}|\boldsymbol{X}, \boldsymbol{\theta}\right)$. This is the case when values of parameters $\boldsymbol{\alpha}$ used in data generating process are greater or equal to the

---

[1]Implementation can be found at https://github.com/andrijaster/GCRFBC_B_NB

[2]PyStruct package does not have option of returning SSVM and CRF confidence values for AUC evaluation

values of parameters $\boldsymbol{\beta}$. This means that the information provided by unstructured predictors is more important for classifications task than the information provided by output structure. Therefore, conditional distribution $P(\boldsymbol{y}, \boldsymbol{z}|\boldsymbol{x}, \boldsymbol{\theta})$ is concentrated around mean value and MAP estimate is a satisfactory approximation. However, when data is generated from distribution with significantly higher values of $\beta$ than $\boldsymbol{\alpha}$, the GCRFBCb performs significantly better than GCRFBCnb. For the larger values of variance norm, this difference is also large. This means that the structure between outputs has significant contribution to solving the classification task. It can be concluded that GCRFBCb has at least equal prediction performance as GCRFBCnb. Also, it can be argued that the models were generally able to utilize most of the information (from both features and the structure between outputs), which can be seen through AUC values. In addition, distribution of local variational parameters were analyzed during learning. It is noticed that in each epoch, the variance of this distribution is small and that the parameters can be clustered and their number significantly reduced. Therefore, it is possible to significantly lower down computational and memory costs of GCRFBCb learning procedure, but that's out of the scope of this paper.

## 4.2 PERFORMANCE ON REAL-WORLD DATASETS

### 4.2.1 SKI LIFTS CONGESTION

Data used in this research includes information on ski lift gate entrances in Kopaonik ski resorts, for the period March 15 to March 30 for the seasons from 2006 to 2011. The goal is to predict occurrence of crowding on ski lifts for 40 minutes in advance. Total number of instances in dataset was 4,850 for each ski lift, which is 33,950 in total.

Relatively simple method for crowding detection was devised for labelling data. We assume that, if the crowding at some gate occurs, distributions of skiing times from other gates to that gate within some time window get shifted towards larger values. We model probability distribution of skiing time between two gates by the well-known parametric method of kernel density estimation (KDE) (Silverman, 2018). The distribution shift is measured with respect to the mode of the distribution. The dataset is generated by observing shifts in time windows of 5 minutes. When the mode of the distribution of skiing times within that window is greater than the mode for the whole time-span, the instance is labeled by 1 (crowding) and otherwise, it is labeled by 0 (no crowding). In order to obtain more information from the data distribution, additional 18 features were extracted.

Four different unstructured predictors that were trained on each class separately were used: ridge logistic regression, LASSO logistic regression, neural network and random forest, whereas additional two unstructured predictors: decision tree and neural network were trained on all nodes together. Additionally, three structural support vector machine and two CRFs classifiers were used (Müller & Behnke, 2014). Fully connected graph of SSVM and CRF models are defined as SSVM-full and CRF-full, whereas Chow-Liu tree method for specifying edge connections are defined as SSVM-tree and CRF-tree, respectively. In the SSVM-independent model the nodes of the graph are not connected.

Six different weighted graphs were used to capture dependence structure between ski lifts (nodes):$\chi^2$ statistics on labels of training set, mutual information between labels, correlation matrix between outputs of over-fitted neural networks, norm of difference between vectors of labels and two graphs were defined based on difference of vectors of historical labels and on differences of historical averages of skier times.

The AUC score and ACC of structured and unstructured predictors, along with the total computational time are shown in Table 1. It can be observed that GCRFBCb and GCRBCnb outperformed unstructured and other structured predictors in all cases. Based on evaluated parameters it could be concluded that dependence structure has significant impact on overall prediction performance, even though, due to low values of norm of variance, GCRFBCb and GCRFBCnb have equal AUC scores. It can be summarized that advantages of structured models compared to unstructured are obvious, but in this particular task due to equal prediction performance and its lower computational and memory complexity, GCRFBCnb is the best choice for this specific application.

Table 1: Prediction performance and computation time of classifiers - Ski lifts congestion problem

| Model | AUC | ACC | Calculation time [sec] |
|---|---|---|---|
| GCRFBCnb | **0.831** | **0.749** | 119.554 |
| GCRFBCb | **0.831** | 0.749 | 3364.326 |
| Ridge logistic | 0.793 | 0.736 | 0.41 |
| LASSO logistic | 0.793 | 0.735 | 1.799 |
| Neural network | 0.790 | 0.720 | 151.571 |
| Random forest | 0.783 | 0.720 | 7.983 |
| Decision tree - together | - | 0.681 | 8.297 |
| Neural network - together | - | 0.711 | 13.997 |
| SSVM - full | - | 0.622 | 517.412 |
| SSVM - tree | - | 0.615 | 580.475 |
| SSVM - independent | - | 0.635 | 1029.172 |
| CRF - tree | - | 0.745 | 16415.723 |
| CRF - full | | 0.740 | 13942.542 |

Table 2: Prediction performance and computation time of classifiers - Music classification accordint to emotion

| Model | AUC | ACC | Calculation time [sec] |
|---|---|---|---|
| GCRFBCnb | 0.859 | 0.811 | 7.248 |
| GCRFBCb | **0.860** | **0.813** | 353.328 |
| Ridge logistic | 0.826 | 0.794 | 0.138 |
| LASSO logistic | 0.832 | 0.797 | 0.874 |
| Neural network | 0.811 | 0.783 | 98.132 |
| Random forest | 0.843 | 0.798 | 2.469 |
| Decision tree - together | - | 0.736 | 0.564 |
| Neural network - together | - | 0.782 | 8.471 |
| SSVM - full | - | 0.755 | 76.817 |
| SSVM - tree | - | 0.795 | 75.93 |
| SSVM - independent | - | 0.784 | 146.867 |

### 4.2.2 MULTI-LABEL CLASSIFICATION OF MUSIC ACCORDING TO EMOTION

The dataset used for this work consists of 100 songs from 7 different genres. The collection was created from 233 musical albums choosing three songs from each album. 8 rhythmic and 64 timbre features are extracted. The music is labeled in 6 categories of emotions: amazed-surprised, happy-pleased, relaxing-calm, quiet-still, sad-lonely and angry-fearful (Trohidis et al., 2008). Total number of instances in dataset was 593. Four different weighted graphs were used: statistics on labels of training set, mutual information between labels, correlation matrix between outputs of over-fitted neural networks and norm of difference between vectors of labels. Same unstructured predictors as in ski lift congestion problem were used, along with three structural support vector machine classifiers.

The performances of models are evaluated by 10 fold cross validation. The AUC score and ACC of structured and unstructured predictors, along with the total computational time are shown in Table 2. It can be seen that GCRFBCb has achieved the best prediction performances. The ACC of GCRFBC models are significantly better than the SSVM performances. The AUC score and ACC of GCRGBCb are higher than the best result (AUC = 0.8237) presented in original paper (Trohidis et al., 2008). As in previous cases, computational time of GCRFBCb is significantly longer compared to GCRFBCnb and SSVM models.

### 4.2.3 GENE FUNCTION CLASSIFICATION

This dataset is formed by micro-array expression data and phylogenetic profiles with 2417 genes (instances). The number of features is 103, whereas each gene is associated with the set of 14 groups (Elisseeff & Weston, 2002). The same unstructured, structured predictors and weighted graphs, as

Table 3: Prediction performance and computation time of classifiers - Gene classification problem

| Model | AUC | ACC | Calculation time [sec] |
|---|---|---|---|
| GCRFBCnb | 0.775 | 0.766 | 48.167 |
| GCRFBCb | **0.797** | **0.775** | 2297.727 |
| Ridge logistic | 0.582 | 0.539 | 0.079 |
| LASSO logistic | 0.583 | 0.540 | 0.188 |
| Neural network | 0.580 | 0.567 | 70.298 |
| Random forest | 0.601 | 0.615 | 5.529 |
| Decision tree - together | - | 0.691 | 1.218 |
| Neural network - together | - | **0.775** | 28.381 |
| SSVM - full | - | 0.771 | 10137.049 |
| SSVM - tree | - | 0.768 | 722.156 |
| SSVM - independent | - | 0.539 | 78.8870 |

in music according to emotion classification, were used. The 10-fold cross validation results of the classification are shown in Table 3.

It can be observed that both GCRFBCb and GCRFBCnb achieved significantly better results in comparison with unstructured predictors. However, neural network trained on all data together achieved the same ACC scores as GCRFBCb. The AUC of GCRFBCb has outperformed Random forest classifier by 19%, whereas SSVM - tree has better ACC compared to GCRFBCnb. It also outperformed GCRFCnb, but as expected, its computation time was longer. In addition, the computation time of CRFs models are longer compared to GCRFBCb

### 4.2.4 HIGHWAY CONGESTION

The E70-E75 motorway is a major transit motorway in Serbia. With 504 kilometers, it is the one the major transit motorway in Serbia. It crosses the country from north-west to south, starting at Batrovci border crossing with the Republic of Croatia and ending with Preševo border crossing with the Republic of North Macedonia.

One of the biggest problems in E70-E75 motorway is high congestion that frequently occurs. One of the reasons lies in lack of open toll stations. In order to mitigate congestion problem, it is necessary to predict its occurrence and open enough toll stations. Data used in this research includes information of car entrance and exit for the year 2017. Two different sections were analyzed: Belgrade - Adaševci and Niš - Belgrade. The section Belgrade - Adaševci was analyzed for the period of January 2017, whereas section Niš - Belgrade was analyzed for the period of April - July 2017. The congestion was labeled using the similar technique based on KDE as presented in the ski lifts congestion problem. Based on raw datasets for sections Niš - Belgrade and Belgrade - Adaševci with 5,132,918 and 487,767 instances, respectively, a new dataset for section Niš - Belgrade is generated by observing shifts in time windows of 10 minutes due to large number of vehicles, whereas in the case of section Belgrade - Adaševci the shifts are observed in time windows of 20 minutes. Total numbers of instances for sections Belgrade - Adaševci and Niš - Belgrade are 50,964 and 235,872, whereas numbers of highway exits (outputs) are 6 and 18, respectively. The extracted features are similar to the ones presented in ski congestion problem. The $\chi^2$ statistics, mutual information, correlation matrix and difference of vectors of historical labels were used to capture dependence structure, whereas the same unstructured predictors as in ski lifts congestion problem were evaluated. The classification results, validated by 10 fold cross validation, are presented in Table 4.

The GCRFBCnb achieved the highest AUC and ACC scores in the section Belgrade - Adaševci, whereas GCRFBCb has better prediction performance in section Niš - Belgrade. Moreover, in case of section Niš - Belgrade, GCRFBCb has worse ACC score than fully connected CRF, whereas CRF-tree outperformed GCRFBCnb in section Belgrade - Adaševci

Table 4: Prediction performance and computation time of classifiers - Highway congestion problem

| | Niš - Belgrade | | | Belgrade - Adaševci | | |
|---|---|---|---|---|---|---|
| | AUC | ACC | Calculation time [sec] | AUC | ACC | Calculation time [sec] |
| GCRFBCnb | 0.740 | 0.684 | 344.166 | **0.974** | **0.925** | 90.321 |
| GCRFBCb | **0.751** | **0.692** | 13818.874 | 0.956 | 0.895 | 2103.749 |
| Ridge logistic | 0.716 | 0.681 | 10.73 | 0.917 | 0.856 | 1.771 |
| LASSO logistic | 0.716 | 0.680 | 30.12 | 0.917 | 0.856 | 1.657 |
| Neural network | 0.72 | 0.682 | 857.602 | 0.956 | 0.904 | 125.339 |
| Random forest | 0.739 | 0.683 | 209.589 | 0.965 | 0.914 | 3.826 |
| Decision tree - together | - | 0.625 | 635.464 | - | 0.898 | 1.893 |
| Neural network - together | - | 0.664 | 125.441 | - | 0.880 | 16.475 |
| SSVM - full | - | 0.588 | 7637.794 | - | 0.739 | 340.806 |
| SSVM - tree | - | 0.588 | 3684.138 | - | 0.755 | 392.597 |
| SSVM - independent | - | 0.602 | 3262.208 | - | 0.814 | 704.07 |
| CRF - tree | - | 0.685 | 29749.054 | - | 0.88 | 26539.250 |
| CRF - full | - | 0.683 | 52563.972 | - | 0.898 | 25339.97 |

## 5 CONCLUSION

In this paper, a new model, called Gaussian Conditional Random Fields for Binary Classification (GCRFBC) is presented. The model is based on latent GCRF structure, which means that intractable structured classification problem can become tractable and efficiently solved. Moreover, the improvements previously applied to regression GCRF can be easily extended to GCRFBC. Two different variants of GCRFBC were derived: GCRFBCb and GCRFBCnb. Empirical Bayes (marginalization of latent variables) by local variational methods is used in optimization procedure of GCFRBCb, whereas MAP estimate of latent variables is applied in GCRFBCnb. Based on presented methodology and obtained experimental results on synthetic and real-world datasets it can be concluded that both GCRFBCb and GCRFBCnb models have better prediction performance compared to the analysed structured unstructured predictors. Additionaly, GCRFBCb has better performance considering AUC score, ACC and lower bound of conditional log likelihood $\mathcal{L}\left(\boldsymbol{Y}|\boldsymbol{X},\boldsymbol{\theta}\right)$ compared to GCRFBCnb, in cases where norm of the variances of latent variables is high. However, in cases where norm of the variances is close to zero, both models have equal prediction performance. Due to high memory and computational complexity of GCRFBCb compared to GCRFBCnb, in cases where norm of the variances is close to zero, it is reasonable to use GCRFBCnb. Additionally, the trade off between complexity and accuracy can be made in situation where norm of the variances is high. Further studies should address extending GCRFBC to structured multi-label classification problems, and lower computational complexity of GCRFBCb by considering efficient approximations.

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

## A  DERIVATION OF LOWER BOUND OF CONDITIONAL LIKELIHOOD

In this section we derive lower bound of conditional likelihood. In order to obtain suitable form of joint distribution that can be easily integrated, the lower bound for sigmoid function was used (Jaakkola & Jordan, 2000). The lower bound of joint distribution $P(\boldsymbol{y}_j, \boldsymbol{z}_j | \boldsymbol{x}_j, \boldsymbol{\theta})$ can be expressed as:

$$P(\boldsymbol{y}_j, \boldsymbol{z}_j | \boldsymbol{x}_j, \boldsymbol{\theta}) = P(\boldsymbol{y}_j | \boldsymbol{z}_j) P(\boldsymbol{z}_j | \boldsymbol{x}_j, \boldsymbol{\theta}) \geq \underline{P}(\boldsymbol{y}_j, \boldsymbol{z}_j | \boldsymbol{x}_j, \boldsymbol{\theta}, \boldsymbol{\xi}_j) \tag{22}$$

$$\underline{P}(\boldsymbol{y}_j, \boldsymbol{z}_j | \boldsymbol{x}_j, \boldsymbol{\theta}, \boldsymbol{\xi}_j) = \prod_{i=1}^{N} \sigma(\xi_{ji}) \exp\left( z_{ji} y_{ji} - \frac{z_{ji} + \xi_{ji}}{2} - \lambda(\xi_{ji})(z_{ji}^2 - \xi_{ji}^2) \right) \cdot \\ \frac{1}{(2\pi)^{N/2} |\Sigma_j|^{1/2}} \exp\left( -\frac{1}{2}(\boldsymbol{z}_j - \boldsymbol{\mu}_j)^T \Sigma_j^{-1} (\boldsymbol{z}_j - \boldsymbol{\mu}_j) \right) \tag{23}$$

The simplified form of Eq. 23 can be represented by rearranging terms in the following form:

$$\underline{P}(\boldsymbol{y}_j, \boldsymbol{z}_j | \boldsymbol{x}_j, \boldsymbol{\theta}, \boldsymbol{\xi}_j) = \mathcal{T}(\boldsymbol{\xi}_j) \exp\left( \boldsymbol{z}_j^T (\boldsymbol{y}_j - \frac{1}{2}\boldsymbol{I}) - \lambda \boldsymbol{z}_j^T \boldsymbol{z}_j - \frac{1}{2}\boldsymbol{z}_j^T \Sigma_j^{-1} \boldsymbol{z}_j + \boldsymbol{z}_j^T \Sigma_j^{-1} \mu \right) \tag{24}$$

$$\mathcal{T}(\boldsymbol{\xi}_j) = \frac{1}{(2\pi)^{N/2} |\Sigma_j|^{1/2}} \prod_{i=1}^{N} \sigma(\xi_{ji}) \exp\left( -\frac{1}{2}\boldsymbol{\mu}_j^T \Sigma_j^{-1} \boldsymbol{\mu}_j - \frac{\xi_{ji}}{2} + \lambda(\xi_{ji})\xi_{ji}^2 \right) \tag{25}$$

The lower bound of likelihood $\underline{P}(\boldsymbol{y}_j | \boldsymbol{x}_j, \boldsymbol{\theta}, \boldsymbol{\xi}_j)$ can be obtained by marginalization of $\boldsymbol{z}_j$ as:

$$\begin{aligned} \underline{P}(\boldsymbol{y}_j | \boldsymbol{x}_j, \boldsymbol{\theta}, \boldsymbol{\xi}_j) &= \int \underline{P}(\boldsymbol{y}_j, \boldsymbol{z}_j | \boldsymbol{x}_j, \boldsymbol{\theta}, \boldsymbol{\xi}_j) d\boldsymbol{z}_j \\ &= \mathcal{T}(\boldsymbol{\xi}_j) \int \exp\left( \boldsymbol{z}_j^T (\boldsymbol{y}_j - \frac{1}{2}\boldsymbol{I}) - \Lambda_j \boldsymbol{z}_j^T \boldsymbol{z}_j - \frac{1}{2}\boldsymbol{z}_j^T \Sigma_j^{-1} \boldsymbol{z}_j + \boldsymbol{z}_j^T \Sigma_j^{-1} \boldsymbol{\mu}_j \right) d\boldsymbol{z}_j \\ &= \mathcal{T}(\boldsymbol{\xi}_j) \int \exp\left( -\frac{1}{2}\boldsymbol{z}_j^T (\Sigma_j^{-1} + 2\Lambda_j) \boldsymbol{z}_j + \right. \\ &\qquad \left. \boldsymbol{z}_j^T (\Sigma_j^{-1} + 2\Lambda_j)(\Sigma_j^{-1} + 2\Lambda_j)^{-1}((\boldsymbol{y}_j - \frac{1}{2}\boldsymbol{I}) + \Sigma_j^{-1}\boldsymbol{\mu}_j) \right) d\boldsymbol{z}_j \end{aligned} \tag{26}$$

The lower bound of likelihood $\underline{P}(\boldsymbol{y}_j | \boldsymbol{x}_j, \boldsymbol{\theta}, \boldsymbol{\xi}_j)$ can be transformed in the following form:

$$\begin{aligned} \underline{P}(\boldsymbol{y}_j | \boldsymbol{x}_j, \boldsymbol{\theta}, \boldsymbol{\xi}_j) &= \mathcal{T}(\boldsymbol{\xi}_j) \int \exp\left( -\frac{1}{2}(\boldsymbol{z}_j - \boldsymbol{m}_j)^T S_j^{-1}(\boldsymbol{z}_j - \boldsymbol{m}_j) + \frac{1}{2}\boldsymbol{m}_j^T S_j^{-1}\boldsymbol{m}_j \right) d\boldsymbol{z}_j \\ &= \mathcal{T}(\boldsymbol{\xi}_j) \exp\left( \frac{1}{2}\boldsymbol{m}_j^T S_j^{-1}\boldsymbol{m}_j \right) \int \exp\left( -\frac{1}{2}(\boldsymbol{z}_j - \boldsymbol{m}_j)^T S_j^{-1}(\boldsymbol{z}_j - \boldsymbol{m}_j) \right) d\boldsymbol{z}_j \end{aligned} \tag{27}$$

where $S_j^{-1} = \Sigma_j^{-1} + 2\Lambda_j$ and $\boldsymbol{m}_j = \Sigma_j \left( (\boldsymbol{y}_j - \frac{1}{2}\boldsymbol{I}) + \Sigma_j^{-1}\boldsymbol{\mu}_j \right)$.

This integration is easily performed by noting that it is the integral over an unnormalized Gaussian distribution, which yields:

$$\underline{P}(\boldsymbol{y}_j | \boldsymbol{x}_j, \boldsymbol{\theta}, \boldsymbol{\xi}_j) = (2\pi)^{N/2} |\Sigma_j|^{1/2} \mathcal{T}(\boldsymbol{\xi}_j) \exp\left( \frac{1}{2}\boldsymbol{m}_j^T S^{-1}\boldsymbol{m}_j \right) |S_j|^{1/2} \tag{28}$$

The final form of the lower bound of conditional log likelihood $\underline{\mathcal{L}}_j(\boldsymbol{y}_j | \boldsymbol{x}_j, \boldsymbol{\theta}, \boldsymbol{\xi}_j)$ is:

$$\underline{\mathcal{L}}_j(\boldsymbol{y_j}|\boldsymbol{x_j}, \boldsymbol{\theta}, \boldsymbol{\xi_j}) = \log \underline{P}(\boldsymbol{y_j}|\boldsymbol{x_j}, \boldsymbol{\theta}, \boldsymbol{\xi_j}) = \sum_{i=1}^{N} \left( \log \sigma(\xi_{ji}) - \frac{\xi_{ji}}{2} + \lambda(\xi_{ji})\xi_{ji}^2 \right) - \\ \frac{1}{2}\boldsymbol{\mu_j}^T \Sigma_j^{-1} \boldsymbol{\mu_j} + \frac{1}{2}\boldsymbol{m_j}^T S_j^{-1} \boldsymbol{m_j} + \frac{1}{2}\log|S_j| \tag{29}$$

# B PARTIAL DERIVATIVE OF LOWER BOUND OF CONDITIONAL LOG LIKELIHOOD

The partial derivative of lower bound of conditional log likelihood (GCRFBCb) $\frac{\partial \underline{\mathcal{L}}_j(\boldsymbol{y_j}|\boldsymbol{x_j}, \boldsymbol{\theta}, \boldsymbol{\xi_j})}{\partial \alpha_k}$ is computed as:

$$\frac{\partial \underline{\mathcal{L}}_j(\boldsymbol{y_j}|\boldsymbol{x_j}, \boldsymbol{\theta}, \boldsymbol{\xi_j})}{\partial \alpha_k} = -\frac{1}{2}\text{Tr}\left( S_j \frac{\partial S_j^{-1}}{\partial \alpha_k} \right) + \frac{\partial \boldsymbol{m_j}^T}{\partial \alpha_k} S_j^{-1} \boldsymbol{m_j} + \frac{1}{2}\boldsymbol{m_j}^T \frac{\partial S_j^{-1}}{\partial \alpha_k} \boldsymbol{m_j} \\ -\frac{\boldsymbol{\mu_j}^T}{\partial \alpha_k} \Sigma_j^{-1} \boldsymbol{\mu_j} - \frac{1}{2}\boldsymbol{\mu_j}^T \frac{\partial \Sigma_j^{-1}}{\partial \alpha_k} + \frac{1}{2}\text{Tr}\left( \Sigma_j \frac{\partial \Sigma_j^{-1}}{\partial \alpha_k} \right) \tag{30}$$

where:

$$\frac{\partial S_j^{-1}}{\partial \alpha_k} = \frac{\partial \Sigma_j^{-1}}{\partial \alpha_k} = \begin{cases} 2, \text{ if } i = j \\ 0, \text{ if } i \neq j \end{cases} \tag{31}$$

$$\frac{\partial \boldsymbol{m_j^T}}{\partial \alpha_k} = -\left( \boldsymbol{y_j} - \frac{1}{2}\boldsymbol{I} + \boldsymbol{\mu_j}^T \Sigma_j^{-1} \right) S_j \frac{\partial S_j^{-1}}{\partial \alpha_k} S_j + \frac{\partial \boldsymbol{\mu_j}^T}{\partial \alpha_k} \Sigma_j^{-1} S_j + \boldsymbol{\mu_j}^T \frac{\partial \Sigma_j^{-1}}{\alpha_k} S_j \tag{32}$$

$$\frac{\partial \mu_j^T}{\partial \alpha_k} = \left( 2\alpha_k R_k(\boldsymbol{x}) - \frac{\partial \Sigma_j^{-1}}{\partial \alpha_k} \boldsymbol{\mu_j} \right)^T \Sigma_j^T \tag{33}$$

Similarly partial derivatives with respect to $\boldsymbol{\beta}$ can be defined as:

$$\frac{\partial \underline{\mathcal{L}}_j(\boldsymbol{y_j}|\boldsymbol{x_j}, \boldsymbol{\theta}, \boldsymbol{\xi_j})}{\partial \beta_l} = -\frac{1}{2}\text{Tr}\left( S_j \frac{\partial S_j^{-1}}{\partial \beta_l} \right) + \frac{\partial \boldsymbol{m_j}^T}{\partial \beta_l} S_j^{-1} \boldsymbol{m_j} + \frac{1}{2}\boldsymbol{m_j}^T \frac{\partial S_j^{-1}}{\partial \beta_l} \boldsymbol{m_j} \\ -\frac{\boldsymbol{\mu_j}^T}{\partial \beta_l} \Sigma_j^{-1} \boldsymbol{\mu_j} - \frac{1}{2}\boldsymbol{\mu_j}^T \frac{\partial \Sigma_j^{-1}}{\partial \beta_l} + \frac{1}{2}\text{Tr}\left( \Sigma_j \frac{\partial \Sigma_j^{-1}}{\partial \beta_l} \right) \tag{34}$$

where:

$$\frac{\partial S_j^{-1}}{\partial \beta_l} = \frac{\partial \Sigma_j^{-1}}{\partial \beta_l} = \begin{cases} \sum_{n=1}^{N} e_{in}^l S_{in}^l(x), \text{ if } i = j \\ -e_{ij}^l S_{ij}^l(x), \text{ if } i \neq j \end{cases} \tag{35}$$

$$\frac{\partial \boldsymbol{m_j^T}}{\partial \beta_l} = -\left( \boldsymbol{y_j} - \frac{1}{2}\boldsymbol{I} + \boldsymbol{\mu_j}^T \Sigma_j^{-1} \right) S_j \frac{\partial S_j^{-1}}{\partial \beta_l} S_j + \frac{\partial \boldsymbol{\mu_j}^T}{\partial \beta_l} \Sigma_j^{-1} S_j + \boldsymbol{\mu_j}^T \frac{\partial \Sigma_j^{-1}}{\beta_l} S_j \tag{36}$$

$$\frac{\partial \mu_j^T}{\partial \beta_l} = \left( -\frac{\partial \Sigma_j^{-1}}{\partial \beta_l} \boldsymbol{\mu_j} \right)^T \Sigma_j^T \tag{37}$$

In the same manner partial derivatives of conditional log likelihood with respect to $\xi_{ji}$ are:

$$\frac{\partial \underline{\mathcal{L}}_j(\boldsymbol{y_j}|\boldsymbol{x_j}, \boldsymbol{\theta}, \boldsymbol{\xi_j})}{\partial \xi_{ji}} = -\frac{1}{2}\text{Tr}\left( 2S_j \frac{\partial \Lambda_j}{\partial \xi_{ji}} \right) - \left[ 2\left( \boldsymbol{y_j} - \frac{1}{2}\boldsymbol{I} \right) S_j \frac{\partial \Lambda_j}{\partial \xi_{ji}} S_j \right] S_j^{-1} \boldsymbol{m_j} \\ + \boldsymbol{m_j}^T \frac{\partial \Lambda_j}{\partial \xi_{ji}} \boldsymbol{m_j} + \sum_{i=1}^{N} \left( \left( \frac{1}{\sigma(\xi_{ji})} + \frac{1}{2}\xi_{ji} \right) \frac{\partial \sigma(\xi_{ji})}{\partial \xi_{ji}} + \frac{1}{2}\left( \sigma(\xi_{ji}) - \frac{3}{4} \right) \right) \tag{38}$$

where:

$$\frac{\partial \Lambda_j}{\partial \xi_{ji}} = \begin{bmatrix} 0 & 0 & 0 & \dots & 0 \\ \vdots & \ddots & \vdots & \ddots & \vdots \\ 0 & 0 & \frac{\partial \lambda(\xi_{ji})}{\partial \xi_{ji}} & \dots & 0 \\ \vdots & \vdots & \vdots & \ddots & \vdots \\ 0 & 0 & 0 & \dots & 0 \end{bmatrix} \tag{39}$$

$$\frac{\partial \sigma(\xi_{ji})}{\partial \xi_{ij}} = \sigma(\xi_{ji})(1 - \sigma(\xi_{ji})) \tag{40}$$

$$\frac{\partial \lambda(\xi_{ji})}{\partial \xi_{ji}} = \frac{1}{2\xi_{ji}} \frac{\partial \sigma(\xi_{ji})}{\partial \xi_{ji}} - \frac{1}{2} \left( \sigma(\xi_{ji}) - \frac{1}{2} \right) \frac{1}{\xi_{ji}^2} \tag{41}$$

## C  PARTIAL DERIVATIVE OF CONDITIONAL LOG LIKELIHOOD

The derivatives of the conditional log likelihood (GCRFBCnb) with respect to $\boldsymbol{\alpha}$ and $\boldsymbol{\beta}$ are defined as, respectively:

$$\frac{\partial \mathcal{L}_{ji}(y_{ji}|\boldsymbol{x_j}, \boldsymbol{\theta}, \mu_{ji})}{\partial \alpha_k} = \left( y_{ji} - \sigma(\mu_{ji}) \right) \frac{\partial \mu_{ji}}{\partial \alpha_k} \tag{42}$$

$$\frac{\partial \mathcal{L}_{ji}(y_{ji}|\boldsymbol{x_j}, \boldsymbol{\theta}, \mu_{ji})}{\partial \alpha_l} = \left( y_{ji} - \sigma(\mu_{ji}) \right) \frac{\partial \mu_{ji}}{\partial \beta_l} \tag{43}$$

where $\frac{\partial \mu_{ji}}{\partial \alpha_k}$ and $\frac{\partial \mu_{ji}}{\partial \beta_l}$ are elements of the vectors $\frac{\partial \boldsymbol{\mu_j}}{\partial \alpha_k}$ and $\frac{\partial \boldsymbol{\mu_j}}{\partial \beta_l}$ and can be obtained by Eqs. 33 and 37, respectively.

## D  SYNTHETIC DATASET RESULTS

In order to generate and label graph nodes, edge weights $S$ and unstructured predictor values $R$ were randomly generated from uniform distribution. Besides, it was necessary to choose values of parameters $\boldsymbol{\alpha}$ and $\boldsymbol{\beta}$. Greater values of $\boldsymbol{\alpha}$ indicate that the model is more confident about performance of unstructured predictors, whereas for the larger value of $\boldsymbol{\beta}$ the model is putting more emphasis on the dependence structure of output variables.

Six different values of parameters $\boldsymbol{\alpha}$ and $\boldsymbol{\beta}$ were used. In the first group $\boldsymbol{\alpha}$ and $\boldsymbol{\beta}$ have similar values, so unstructured predictors and dependence structure between outputs have similar importance. In the second group, $\boldsymbol{\alpha}$ has higher values compared to $\boldsymbol{\beta}$, which means that unstructured predictors are more important than the dependence structure. In the third group $\boldsymbol{\beta}$ has higher values than $\boldsymbol{\alpha}$, meaning that dependence structure is more important than unstructured predictors.

Along with the AUC and conditional log likelihood, norm of the variances of latent variables (diagonal elements in the covariance matrix) is evaluated and presented in Table 5. In addition, the results of experiments are presented in Fig. 1, where for different values of $\boldsymbol{\alpha}$ and $\boldsymbol{\beta}$ we show differences between GCRFBCb and GCRFBCnb (a) AUC scores, (b) log likelihoods, and (c) norm of the variances of latent variables.

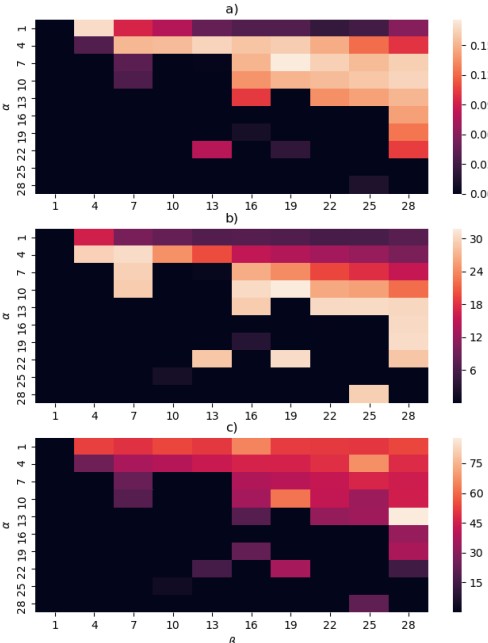

Figure 1: Experimental evaluation of differences between GCRFBCb and GCRFBCnb (a) AUC scores, (b) log likelihoods, and (c) norms of the variances of latent variables for different values of $\boldsymbol{\alpha}$ and $\boldsymbol{\beta}$

Table 5: Comparison of GCRFBCb and GCRFBCnb prediction performance for different values of $\boldsymbol{\alpha}$ and $\boldsymbol{\beta}$, as measured by AUC, log likelihood, and norm of diagonal elements of the covariance matrix

| No. | Parameters | GCRFBCb | | | GCRFBCnb | |
|---|---|---|---|---|---|---|
| | | AUC | $\mathcal{L}\left(\boldsymbol{Y}\|\boldsymbol{X},\boldsymbol{\theta}\right)$ | $\|\boldsymbol{\sigma}\|_2$ | AUC | $\mathcal{L}\left(\boldsymbol{Y}\|\boldsymbol{X},\boldsymbol{\theta}\right)$ |
| **1** | $\boldsymbol{\alpha}=[5,4]$ $\boldsymbol{\beta}=[5,22]$ | 0.812 | -71.150 | 0.000 | 0.812 | -71.151 |
| **2** | $\boldsymbol{\alpha}=[1,18]$ $\boldsymbol{\beta}=[1,18]$ | 0.903 | -75.033 | 0.001 | 0.902 | -75.033 |
| **3** | $\boldsymbol{\alpha}=[22,21]$ $\boldsymbol{\beta}=[5,22]$ | 0.988 | -83.957 | 0.000 | 0.988 | -83.957 |
| **4** | $\boldsymbol{\alpha}=[22,21]$ $\boldsymbol{\beta}=[0.1,0.67]$ | 0.866 | -83.724 | 0.000 | 0.886 | -83.466 |
| **5** | $\boldsymbol{\alpha}=[0.8,0.5]$ $\boldsymbol{\beta}=[5,22]$ | 0.860 | -83.353 | 34.827 | 0.817 | -84.009 |
| **6** | $\boldsymbol{\alpha}=[0.2,0.4]$ $\boldsymbol{\beta}=[1,18]$ | 0.931 | -70.692 | 35.754 | 0.821 | -70.391 |

