# OpenReview forum: "Gaussian Conditional Random Fields for Classification"
_ICLR.cc/2020/Conference — Reject_

### Official Review · AnonReviewer1 · 2019-10-21
**Official Blind Review #1**

**Rating:** 1

**Review:**

The work involves modifiying gaussian conditional random fields to work for classification problems instead of regression problems. The main idea is to apply a bernoulli distribution on top of the regression values to convert them to work with binary classification problems. Two variations are discussed along with the inference and learning methodology. The inference can be done using numerical approximation and learning using variational methods and is still untracktable. Comparisons with other modeling strategies is done using experiments.

The paper is incremental and doesn't really provide improvements to learning parameters (or at least there is no theory showing this in the paper). The experiments do not seem satisfactory as discussed below.
a) Applying a bernoulli distribution on the output of the GCRF seems trivial. It is not very clear when the GCRFBCb model would be better than the GCRFBCnb. The learning procedure is untracktable and hard to follow on why this might provide better results.
b) The datasets (music classification and gene classification) don't seem to be good datasets for structured predictions i.e. the interaction needed between the nodes is not clear. Since they are multilabel problems, one could have just modeled the system with N independent nodes or design a multinomial distribution instead of only for binary classification.
c) There should be more thorough fine-tuning of other models, for e.g. in the ski lifts experiment, the CRF does much worse than logistic regression in the results. This is most likely because the parameters were not initialized properly using normal tricks like using logistic regression. Typically for truly structured problems, CRFs do better than their logistic regression counter parts. It is also not clear how the other models (CRF and SSVM) pairwise potentials were modeled.

It would really help to make this paper stronger by showing the new modeling technique does better than CRFs (that are tuned properly) on better structured datasets. It would be good to have a discussion on when this model would do worse than the other structured models and why.


**Experience Assessment:**

I have published one or two papers in this area.

**Review Assessment: Checking Correctness Of Derivations And Theory:**

I assessed the sensibility of the derivations and theory.

**Review Assessment: Checking Correctness Of Experiments:**

I assessed the sensibility of the experiments.

**Review Assessment: Thoroughness In Paper Reading:**

I read the paper at least twice and used my best judgement in assessing the paper.

---

> ### Author Response · Authors · 2019-11-14
> **Response to Official Blind Review #1**
>
> Point 1. Applying a bernoulli distribution seems trivial.
>
> Applying the Bernoulli distribution on the outputs of the GCRF might seem trivial in the case of GCRFBnb model since the mode of GCRF over latent variables is used in learning and inference, which makes this case straightforward.  However, as it is well known, probabilistic model which does not consider full distributions but only the mode of the distribution loses important information and often does not perform well. Therefore, we introduced latent dependence structure and marginalized the joint distribution of P(Y,Z) over latent variables, which is not an easy task. In order to lower the computation cost we marginalized the distribution using local variation approximation. To the best of our knowledge, this is the first time that local variation approximation was used in the case of several dependent Bernoulli variables.  Additionally because of the model representation we derived the inference procedure which is straightforward and have low computational cost.
>
> Point 2. When the GCRFBCb model would be better than the GCRFBCnb.
>
> In the appendix D we presented in table 5 and figure 2 detailed results obtained on synthetic dataset that can explain where the GCRFBCb model is performing better than the GCRFBCnb. We emphasized in section Experimental evaluation - synthetic dataset, in which cases GCRFBCb performs better than GCRFBCnb.  Comparison of GCRFBCb and GCRFBCnb performances are presented in figure 2.
> It can be noticed that in cases where variances of latent variables are relatively small, both models have equal performance considering AUC and conditional log likelihood.  This means that results obtained by unstructured predictors are equally or more important for classification task compared to the structure between outputs. In such case, MAP estimate is a satisfactory approximation.
> However, when data were generated from a distribution with significantly higher values of $\beta$ compared to $\alpha$, the GCRFBCb performs significantly better than GCRFBCnb. This means that the structure between outputs has significant contribution to classification task compared to the results obtained by unstructured predictors. It can be concluded that GCRFBCb has at least equal prediction performance as GCRFBCnb. and that the models were generally able to utilize most of the information (from both features and the structure between outputs).
>
> Point 3. The learning procedure is untracktable and hard to follow.
>
> Indeed, learning procedure of GCRFBCb is more complex than the one of GCRFBCnb. That is due to the marginalization over latent variables which allows the model to exploit more information than the one relying only on the modes of latent variable distributions in case that the variance of these distributions is large (as discussed in Point 2).  It is also true that the procedure is computationally more demanding, but we wouldn’t qualify it as intractable, since we managed to perform training in reasonable time. Also, learning procedures of structured models are usually computationally more demanding and so is ours. But the computational and conceptual complexity of the procedure paid off in better prediction results as our experimental evaluation shows.
>
> Point 4. The multilabel datasets  don't seem to be good datasets for structured predictions.
>
> We noticed that in many papers that are focused on structured predictions the multilabel problems were given. The multilabel problems can be defined as structured prediction problems, because structure between labels can have significant impact on classification scores. However, in response to your request we added two additional datasets that are completely structured. Both dataset are connected with the highway congestion prediction on two different highways in Europe. Now we demonstrate consistent improvement on 3 completely structured datasets.
>
>
> Point 5. There should be more thorough fine-tuning of other models.
>
> When we considered your suggestion, we noticed that we indeed didn't properly train CRFs, so we rerun the experiments and indeed they now perform better than logistic regression models, but our approach still outperforms them. The pairwise potential that we used in CRF are implemented in pystruct module of CRF (https://pystruct.github.io/user_guide.html).
>
> Point 6. It would be good to have a discussion on when this model would do worse than the others.
>
> Regarding potential drawbacks of our model, it relies on the assumption that the underlying distribution of latent variables is multivariate normal distribution, so in the case when this distribution cannot be fitted well to the data (e.g. when the distribution of latent variables is multimodal), the model won't perform as well as it is expected. In response to your suggestion, now we emphasize this in the paper. But also, compared to classical CRFs and SSVM our model has three important advantages, as it is emphasized in the Introduction.

---

### Official Review · AnonReviewer3 · 2019-10-22
**Official Blind Review #3**

**Rating:** 6

**Review:**

TITLE
Gaussian Conditional Random Fields for Classification

REVIEW SUMMARY
A well justified approach to structured classification with demonstrated good performance.

PAPER SUMMARY
The paper presents methods for structured classification based on a Gaussian conditional random field combined with a softmax Bernoulli likelihood. Methods for inference and parameter learning are presented both for a "Bayesian" and maximum likelihood version. The method is demonstrated on several data sets.

QUALITY
In general, the technical quality of the paper is good. Except for minor typos, derivations appear to be correct, although I did not check everything in detail.

CLARITY
The paper could be improved by a careful revision with focus on improving grammar, but as it stands the paper is easy to follow.
It is not clear to me exactly how the numbers in Table 1 were computed. Is this based on 10-fold crossvalidation as in the following tables?

ORIGINALITY
I am not familiar enough with the field to assess the novelty of the contribution. It would be great if the paper provided a better overview of competing structured classification methods.

FURTHER COMMENTS

"structured classification" ?

"It was shown" -> We show

"for given"

Is the second sum over k=1 to K in eq. 1 a mistake?

"We void" -> We avoid



**Experience Assessment:**

I do not know much about this area.

**Review Assessment: Checking Correctness Of Derivations And Theory:**

I assessed the sensibility of the derivations and theory.

**Review Assessment: Checking Correctness Of Experiments:**

I assessed the sensibility of the experiments.

**Review Assessment: Thoroughness In Paper Reading:**

I made a quick assessment of this paper.

---

> ### Author Response · Authors · 2019-11-14
> **Response to Official Blind Review #3**
>
> Point 1. The paper could be improved by a careful revision with focus on improving grammar, but as it stands the paper is easy to follow.
>
> Thank you for your suggestion we revised the paper and improved the grammar.
>
> Point 2. It is not clear to me exactly how the numbers in Table 1 were computed. Is this based on 10-fold crossvalidation as in the following tables?
>
> Results obtained in all tables are obtained by 10-fold cross validation.
>
> Point 3. It would be great if the paper provided a better overview of competing structured classification methods.
>
> Additional references and discussion concerning relevant references connected with competing structured classification methods and their applications are added in related work.

---

### Official Review · AnonReviewer2 · 2019-10-24
**Official Blind Review #2**

**Rating:** 6

**Review:**

The authors break the double blind anonymity with the code link provided. I'll leave how to deal with this to the meta reviewer.

The authors provide a method to modify GRFs to be used for classification. The idea is simple and easy to get through, the writing is clean. The method boils down to using a latent variable that acts as a "pseudo-regressor" that is passed through a sigmoid for classification. The authors then discuss learning and inference in the proposed model, and propose two different variants that differ on scalability and a bit on performance as well. The idea of using the \xi transformation for the lower bound of the sigmoid was interesting to me -- since I have not seen it before, its possible its commonly used in the field and hopefully the other reviewers can talk more about the novelty here. The empirical results are very promising, which is the main reason I vote for weak acceptance. I think the paper has value, albeit I would say its a bit weak on novelty, and I am not 100% convinced about the this conference being the right fit for this paper. The authors augment MRFs for classification and evaluate and present the results well.

Can the authors intuit why random forests and neural nets dont perform as well ? It seems there are many knobs one can tune to get better performance, so I will take the presented results with a grain of salt. Also, it seems one can also use other "link" functions with MRFs (similar to link functions in generalized linear models) to not just do logistic but other possible losses as well. How about multiclass classification using softmax ? I think such generalizations would make this paper lot stronger.

**Experience Assessment:**

I have read many papers in this area.

**Review Assessment: Checking Correctness Of Derivations And Theory:**

I carefully checked the derivations and theory.

**Review Assessment: Checking Correctness Of Experiments:**

I assessed the sensibility of the experiments.

**Review Assessment: Thoroughness In Paper Reading:**

I read the paper at least twice and used my best judgement in assessing the paper.

---

> ### Author Response · Authors · 2019-11-14
> **Response to Official Blind Review #2**
>
> Point 1. The authors break the double blind anonymity with the code link provided.
>
> Reviewer #2 commented that we broke the double blind review anonymity by providing the link to the code. Please note that the repository does not contain any information on the authors nor their institutions. The link was provided solely for purposes of better evaluation of our work by the reviewers.
>
> Point 2. Can the authors intuit why random forests and neural nets don't perform as well ?
>
> There are two reasons why random forests and neural nets cannot outperform classification accuracy of GCRFBC models.
> 1. The random forests and neural networks are unstructured classifiers, meaning that they model outputs as conditionally independent given inputs, so they do not consider structure between output variables which contains valuable information. In structured (Ski lift congestion dataset and Highway congestion dataset) and multilabel (Gene functional dataset and Music according to emotion dataset) classification tasks information shared among outputs have significant impact on classification accuracy, so unstructured classifiers are outperformed.
> 2. GCRF for classification can also be seen as an ensemble model, which takes the outputs of unstructured classifiers as inputs. Therefore, GCRF for classification is able to figure out how reliable unstructured classifiers are and how much predefined output structure is significant for modeling and that way boost classification accuracy compared to unstructured models.
>
> Point 3. It seems there are many knobs one can tune to get better performance, so I will take the presented results with a grain of salt.
>
> In this revision we payed attention to fine-tune hyperparameters of all classifiers. The results in the paper are updated, but there is no important difference.
>
>
> Point 4. Also, it seems one can also use other "link" functions with MRFs (similar to link functions in generalized linear models) to not just do logistic but other possible losses as well. How about multiclass classification using softmax ?
>
> You are absolutely right. Due to the conditional independence of elements of output vector (yj) given corresponding latent variable (zj) it is possible to use different loss functions. In the case of non Bayesian GCRFBC (GCRFBCnb) the learning procedure and inference is straightforward in multiclass case, too. However, in our experience, marginalization of joint distribution for multiclass classification problem is not easily defined relying on local variational approximation. Therefore, other variational approaches (e.g. variational autoencoders with reparametrization trick) would be preferred, but they require more computational power compared to procedure that is implemented in this paper. However, we intend to explore that approach in our future work.

---

### Author Response · Authors · 2019-11-14
**Feedback Incorporated In New Paper Version**

Thanks to all the reviewers for their helpful and constructive feedback. We have uploaded a new paper revision to address the comments and feedback:

1. Added discussion on advantages and disadvantages of GCRFBC model (Introduction).
2. Additional references and discussion concerning relevant references connected with competing structured classification methods and their applications are added in Related work.
3. Added figure 2 and discussion about GCRFBCb and GCRFBCnb model performance in appendix D.
4. Fixed typos and grammar mistakes.
5. All experiments were updated and hyperparameters were fine-tuned.
6. Experimental evaluation on two new structured datasets (now in total we use 5) concerning highway congestion prediction were added.

Please let us know in case of any additional questions or further suggestions on how the paper can be improved.

---

### Decision · Program_Chairs · 2019-12-19

**Decision:**

Reject

**Comment:**

Main content:

Blind review #2 summarizes it well:

The authors provide a method to modify GRFs to be used for classification. The idea is simple and easy to get through, the writing is clean. The method boils down to using a latent variable that acts as a "pseudo-regressor" that is passed through a sigmoid for classification. The authors then discuss learning and inference in the proposed model, and propose two different variants that differ on scalability and a bit on performance as well. The idea of using the \xi transformation for the lower bound of the sigmoid was interesting to me -- since I have not seen it before, its possible its commonly used in the field and hopefully the other reviewers can talk more about the novelty here. The empirical results are very promising, which is the main reason I vote for weak acceptance. I think the paper has value, albeit I would say its a bit weak on novelty, and I am not 100% convinced about the this conference being the right fit for this paper. The authors augment MRFs for classification and evaluate and present the results well.

--

Discussion:

As blind review #1 points out:

Even from the experiments (including the new traffic one), it is unclear how much better the method is either because we don't know if the improvements are statistically significant and that in many of the results, unstructured models like RF or logistic regression are very competitive casting some doubt on whether these datasets were well suited for structured prediction.

--

This paper is a desk reject as review #2's points out that anonymity was broken by the inclusion of a code link that reveals the authorship, which is true as a simple search on the GitHub user "andrijaster" immediately brings us to https://arxiv.org/pdf/1902.00045.pdf which is a draft of this submission showing all author names.